# A Method of Precise Auto-Calibration in a Micro-Electro-Mechanical System Accelerometer

**DOI:** 10.3390/s24124018

**Published:** 2024-06-20

**Authors:** Sergiusz Łuczak, Magdalena Ekwińska, Daniel Tomaszewski

**Affiliations:** 1Warsaw University of Technology, Faculty of Mechatronics, 00-661 Warsaw, Poland; 2Łukasiewicz Research Network Institute of Microelectronics and Photonics, Al. Lotnikow 32/46, 02-668 Warsaw, Poland; magdalena.ekwinska@imif.lukasiewicz.gov.pl (M.E.); daniel.tomaszewski@imif.lukasiewicz.gov.pl (D.T.)

**Keywords:** MEMS accelerometer, auto-calibration, surface micromachining, tunneling current

## Abstract

A novel design of a MEMS (Micro-Electromechanical System) capacitive accelerometer fabricated by surface micromachining, with a structure enabling precise auto-calibration during operation, is presented. Precise auto-calibration was introduced to ensure more accurate acceleration measurements compared to standard designs. The standard mechanical structure of the accelerometer (seismic mass integrated with elastic suspension and movable plates coupled with fixed plates forming a system of differential sensing capacitors) was equipped with three movable detection electrodes coupled with three fixed electrodes, thus creating three atypical tunneling displacement transducers detecting three specific positions of seismic mass with high precision, enabling the auto-calibration of the accelerometer while it was being operated. Auto-calibration is carried out by recording the accelerometer indication while the seismic mass occupies a specific position, which corresponds to a known value of acting acceleration determined in a pre-calibration process. The diagram and the design of the mechanical structure of the accelerometer, the block diagram of the electronic circuits, and the mathematical relationships used for auto-calibration are presented. The results of the simulation studies related to mechanical and electric phenomena are discussed.

## 1. Introduction

Despite the many advantages of MEMS accelerometers [1], it is well known that one of their weakest points is low accuracy, resulting from, e.g., the instability of their operational parameters, which is caused, for example, by thermal [2] or long-term drifts [3], and the effects of aging of the accelerometer structure [4,5]. Yet another difficulty is the necessity to calibrate MEMS accelerometers before their use if a higher indication accuracy is required, as it is not possible to use the operational parameter values provided in the relevant datasheets, since each accelerometer is characterized by individual values [6]. This is especially evident in the case of low-cost MEMS accelerometers, especially when the calibration process must be performed by the user [2]. On the other hand, in the case of factory-calibrated accelerometers, their price increases, and there is still a risk that the averaged values of factory-calibrated parameters do not match the actual values [7], which degrades the accuracy of the accelerometer indications.

That is why various technical solutions have been developed to improve or facilitate the process of calibrating MEMS accelerometers by the user. Recently, many research teams have dealt with this important issue. Various novel calibration techniques have been proposed, e.g., a lightweight method requiring low computational power [8], a Bayesian method [9], dynamic calibration [10], an improved ellipsoid fitting algorithm [11], a method using pseudo acceleration [12], and autonomous calibrations [13,14]. Moreover, new sensor designs have been developed, e.g., a sensor with digital automatic self-calibration [15] and a sensor with a recalibration feature [16]. However, proposals of a modified mechanical structure of a MEMS accelerometer capable of ensuring autonomous calibration are rarely reported.

The process of calibrating MEMS accelerometers is conventionally carried out by means of expensive laboratory stands, presented, for example, in [17,18]. To eliminate the need to use a test stand, a special housing of a commercial IMU was proposed, which enables the easy calibration of the embedded three-axis MEMS accelerometer and gyroscope [16,19]. A simple calibration of the accelerometer without using laboratory equipment is very advantageous, especially for applications that are related to research fields other than mechanical engineering, e.g., human activity recognition for computational [20], sporting [21], or medical [22,23] purposes.

Another solution in the case of triaxial low-g accelerometers is the use of auto-calibration, proposed a long time ago [24] and is still widely used [25,26,27,28,29,30,31,32], which is based on the fact that under static or quasi-static conditions, the geometric sum of the Cartesian acceleration components, particularly the accelerometer axes, is equal to the gravitational acceleration (with an accuracy resulting from existing errors). However, depending on the accepted procedure, this approach may be characterized by relative errors of one magnitude of even several percent [24]. This is why other concepts for achieving auto-calibration have been proposed, e.g., the application of adaptive filters [33,34].

Yet another approach is to expand the mechanical silicon structure of the accelerometer itself. Accelerometers with built-in temperature sensors, enabling compensation for thermal errors—for example, the ADIS accelerometer series by Analog Devices Inc. [35]—have been available for a long time. It has also been proposed to use external micro-heaters to maintain a constant temperature throughout the entire accelerometer [36,37]. Some companies store averaged characteristics, predicting the effects of accelerometer structure aging in the accelerometer memory in order to eliminate errors related to this [5]. Another suggestion is a use of a built-in “self-test” function, simulating the operation of external acceleration in order to test the correctness of the accelerometer’s response to such acceleration [38]. Unfortunately, the self-test function is not performed with sufficiently high precision, thereby making it unsuitable for accurate accelerometer calibration without some modifications.

This paper presents a method to precisely auto-calibrate a MEMS-type capacitive accelerometer with a modified mechanical silicon structure. The accelerometer may feature an arbitrary number of sensitivity axes provided that it is possible to expand each sensitivity axis with additional position detectors (as discussed later in the text). Calibration consists of recording the output signal of the accelerometer during its standard operation at the very moment the seismic mass occupies a specified position, which is detected with high accuracy owing to the use of additional position detectors. Such detectors can operate, for example, on the principle of generating a tunneling current phenomenon. The high nonlinearity between the tunneling current and the displacement [39] is a huge problem when employed as the basic acceleration-sensing principle. Nevertheless, there are many examples in which a tunneling current transducer, employing a tunneling current passing through the air gap between the tip and the plate [40], has been applied in the structure of a MEMS accelerometer, as presented, e.g., in [41,42,43,44]. A typical solution to this nonlinearity is usually the operation of the accelerometer with force feedback [45].

However, in the case of the proposed detectors, this nonlinearity is a significant advantage since it ensures an extremely high sensitivity over a very limited displacement range [40]. Moreover, an accurate and stable tunneling current transducer is not required, since the specific position can be determined on the basis of the peak of the recorded tunneling current regardless of its absolute value.

Reading the accelerometer output signal associated with a given sensitivity axis at these specific positions enables auto-calibration with very high accuracy. In cases when the seismic mass vibrates (which usually occurs) with a sufficiently large amplitude (not less than, e.g., 50% of the measurement range), auto-calibration can be repeated frequently, and thus, it is possible to eliminate many significant accelerometer indication errors (mainly long-term drifts, thermal drifts, errors due to voltage instability, and aging effects). The construction of the detectors must be adapted to the mechanical structure of a particular accelerometer. Their proper construction will enable the accurate detection of the specific positions in each case despite the dispersion of electrical and mechanical parameters of individual accelerometers in the production batch.

The proposed modification makes it possible to obtain very high accuracy in indications of MEMS accelerometers. Moreover, it makes it possible to eliminate some embedded electronic circuits.

This paper is organized in the following way: Section 2 describes the mechanical design of the accelerometer, Section 3 explains the principle of the auto-calibration, Section 4 explains the electronic structure and the operating principle of the accelerometer, Section 5 presents the mechanical model of the accelerometer, the simulation of displacements of the seismic mass, and the simulation related to the tunneling current transducer, Section 6 discusses the main problems related to the proposed concept and proposes alternative designs of the accelerometer, Section 7 provides a brief summary, and Section 8 provides information about the granted patent.

## 2. The Mechanical Design of the Accelerometer

A standard MEMS accelerometer, modeled according to the concept proposed by Analog Devices Inc., Wilmington, MA, USA [46], implemented, e.g., in the ADXL 105 accelerometer [47], contains a seismic mass in the shape of an elongated element integrated at opposite ends with elastic suspensions, which are permanently bonded to the substrate of the accelerometer at the bonding points (anchors). At the same time, the seismic mass is equipped with a series of movable sensing plates, P1, with each having stationary counterparts on both sides, labeled as sensing plates P2 and P3, thus forming a differential sensing capacitor system, which enables an analogue measurement of the position of the seismic mass and hence the measurement of the linear acceleration acting on it. At the anchors, there is a permanent bond between the substrate and the elastic suspension as well as the stationary sensing plates. The seismic mass and the elastic suspension are above the substrate, and there is an air gap between these elements, which allows the seismic mass to move freely. The structure is shown in Figure 1.

In the proposed modified accelerometer design, the standard electric structure was enhanced by adding four detection electrodes. One of them, E1, is moving as it is attached to the seismic mass, while the other three, E2, E3, and E4, are stationary and are bonded to the substrate of the accelerometer. Together with the moving detection electrode E1, they form three detectors for detecting specific positions of the seismic mass: the central position (detector D2: electrodes E1 and E3, as shown in Figure 1) as well as outer positions left (detector D1: electrodes E1 and E2, as shown in Figure 2a) and right (detector D3: electrodes E1 and E4, as shown in Figure 2b). Note the inverse reaction of the seismic mass with respect to the direction of the acting acceleration due to the inertial force in Figure 2a,b. Each of the detection electrodes, moving and stationary, has an elongated shape and a pointed tip, and as such, they create three identical tunneling current transducers.

It was assumed that the outer specific positions of the seismic mass correspond, for example, to 50% of the measurement range *R* of the accelerometer. The outputs of the detectors of the central and the extreme specific positions of the seismic mass are connected to a dedicated auto-calibration circuit (see Figure 3).

Since the accelerometer will most probably operate under thermal conditions that are different from those under which we pre-determined the acceleration value corresponding to the generation of a peak of the current signal by the detectors of the specific positions of the seismic mass, it is necessary to design the elastic suspension of the seismic mass in such a way that its stiffness is independent of the temperature. This can be achieved by carrying out temperature compensation for this structural element, which consists of the appropriate selection of the shape, dimensions, and material of the elastic suspension of the seismic mass, through which changes in the value of the elastic modulus are compensated by changes in the dimensions of the suspension. The elastic suspension of the seismic mass designed in such a way will be characterized by a constant value of stiffness over a sufficiently wide range of temperature changes.

## 3. The Principle of Auto-Calibration

The proposed idea of self-calibration is based on realizing two procedures: one initial pre-calibration and then repeatable post-calibrations. The first is more significant as it affects the precision of the second. It requires an external reference source, providing accelerations whose magnitude (or component in the sensitive axis of the accelerometer) corresponds to the alignment of the pairs of the detection electrodes (i.e., the generation of an appropriate current signal).

The second procedure is repeated automatically at the moment when one of the pre-determined accelerations acts (in our case, ±50% of the measurement range or 0) and consists of updating values of the scale factor and the offset of the accelerometer.

The pre-calibration procedure should be repeated from time to time if the aging effects are to be eliminated.

Referring to MEMS accelerometers, the measurement of linear acceleration consists of using the following relationship [18]:(1)$$a=\frac{U-O}{S},$$
where

*a*—the measured acceleration;*U*—the output voltage resulting from the instantaneous value of the capacitance of the differential capacitor consisting of plates P1, P2, and P3;*O*—the offset (bias) of the output voltage expressed in Volts [V];*S*—the scale factor of the output voltage expressed in Volts per *g* [V/g];*g*—gravitational acceleration (about 10 m/s^2^).

### 3.1. Pre-Calibration of Specific Acceleration

As a standard procedure, the parameters *O* and *S* are individually determined during the calibration or auto-calibration of a MEMS accelerometer for each sensitive axis [25,26]. Because of manufacturing errors, it is necessary to experimentally determine a precise value of each external acceleration corresponding to the specific positions of the seismic mass (Figure 2a,b), where maximal currents are generated by the respective pair of the detection electrodes. These values are recorded in the non-volatile memory of the accelerometer during pre-calibration, which is performed in a standard way: by using a dedicated test rig that generates acceleration, as proposed in [17], or by using a dedicated test rig that applies a tilt to the accelerometer, as presented in [18,26], when gravitational acceleration is employed. This pre-calibration makes it possible to precisely determine the *w* value—the ratio of the acceleration related to the full measurement range *R* (in our case, the ratio was accepted as 0.5 (50%)). The ratio should then be used while performing successive post-calibrations by using Equations (3) and (6).

### 3.2. Post-Calibration of Accelerometer

The operating principle of the detectors regarding the specific positions of the seismic mass is that they only detect the situation when the moving detection electrode, E1, is very close to the respective stationary detection electrode at E2, E3, or E4. As mentioned above, such a position can be detected using the tunneling current. In other words, the electrodes make it possible to precisely determine three specific positions of the seismic mass corresponding to the action of a pre-determined value of the measured linear acceleration. In cases when the seismic mass is exactly at one of the three specific positions, the accelerometer indication is read by means of the measurement plates P1, P2, or P3, forming the differential measuring capacitor system, and it is stored in the memory.

Based on the recorded indications of the accelerometer at two different specific positions of the seismic mass, it is possible to perform the auto-calibration of the accelerometer, which consists of calculating two key operational parameters: the offset *O* and the scale factor *S* of the output signal. Assuming that detector D2 corresponds to a zero acceleration, and detectors D1 and D3 correspond to an acceleration equal to 50% of the full measurement range *R* of the accelerometer (expressed as a multiple of *g*), the parameters *O* and *S* are calculated according to the below formulas.
When the indication *U*_2_ of the accelerometer, corresponding to the coverage of the E1 and E3 electrodes, is recorded, as shown in Figure 1, and the indication *U*_1_ of the accelerometer, corresponding to the coverage of the E1 and E2 electrodes, is recorded, as shown in Figure 2a, the following is true:
(2)$$O=U_{2},$$
and
(3)$$S=\frac{\left(U_{1}-U_{2}\right)}{w\cdot R}.$$
b.When the indication *U*_2_ of the accelerometer, corresponding to the coverage of the E1 and E3 electrodes, is recorded, as shown in Figure 1, Equation (2) is true, and then the indication *U*_3_ of the accelerometer, corresponding to the coverage of the E1 and E4 electrodes, is recorded as follows, as shown in Figure 2b:
(4)$$S=\frac{\left(U_{2}-U_{3}\right)}{w\cdot R}.$$
c.When the indication *U*_1_ of the accelerometer, corresponding to the coverage of the E1 and E2 electrodes, is recorded, as shown in Figure 2a, then the indication *U*_3_ of the accelerometer, corresponding to the coverage of the E1 and E4 electrodes, is recorded as follows, as shown in Figure 2b:
(5)$$O=\frac{\left(U_{1}+U_{3}\right)}{2},$$
and
(6)$$S=\frac{\left(U_{1}-U_{3}\right)}{2w\cdot R}$$
where *w*—the ratio of the acceleration corresponding to the outer specific positions of the seismic mass to the full measurement range *R*; in our case, the ratio was accepted as 0.5 (50%).

Auto-calibration makes it possible to calibrate the sensitivity axis of the MEMS accelerometer during its operation without any additional devices.

The principle of operation of detectors D1, D2, and D3 of the specific positions of the seismic mass is as follows: When the accelerometer is subjected to a predetermined acceleration of a strict value (e.g., equal to 50% of the measurement range), the given detector generates a peak value of its output signal as a result of the minimum distance between the moving electrode E1 and the corresponding stationary electrode (E2, E3, or E4). As one of the possible solutions, this may be, e.g., the maximum peak value of the tunneling current. At the moment of generating such a signal, the voltage generated by the main electronic transducer of the accelerometer is read and saved in the memory of the auto-calibration system. Then, recursively, using the aforementioned formulas and the values of the appropriate voltages saved in the previous step, values of the *O* and *S* parameters are computed in the auto-calibration circuit and then sent to the signal conditioning and calibration circuit, where they are stored in the memory in order to determine a current value of the measured acceleration with higher precision.

## 4. The Electronic Structure of the Accelerometer

Figure 3 shows a schematic diagram of the electronic layer of the accelerometer. In the main measuring circuit of the accelerometer, the measured acceleration *a* is converted into the displacement of the seismic mass *d*, and then the displacement *d* is converted into the change in the differential capacitance *C*. Then, the capacitance is finally processed and converted into the voltage *U*. The measured acceleration *a* also affects detectors D1, D2, and D3 of the specific positions of the seismic mass, which generate the peak value of their output signal (e.g., current) in a situation where the acceleration assumes a strictly defined value.

Optionally, in order to carry out the auto-calibration process, a modified “self-test” functionality can be activated, provided the voltage supplied to the electrostatic actuator (low-frequency voltage between the sensing electrodes) increases slowly enough to register the peak signal from one of the detectors of the specific position of the seismic mass with high precision. With such a modification, the auto-calibration process can be performed on demand (or automatically) at any time provided that no acceleration acts in a given sensitivity axis, having a value resulting in operation beyond the specific outer positions of the seismic mass, e.g., in our case, greater than 50% of the measurement range.

The electronic structure of the sensor, which is responsible for the standard operation of the sensor, is presented in Figure 4.

## 5. Simulation Study

### 5.1. The Mechanical Structure of the Accelerometer

The mechanical structure of a uniaxial accelerometer was developed for simulation purposes and is presented in Figure 5. The distance between the individual comb teeth and the flat surface is 5 µm. The width of each comb tooth is 2 µm, and its height is 5 µm. The distance between two combs (the distance between the movable teeth and the stationary surface of the second comb) is also 5 µm. Therefore, we can treat the described system as multiple surfaces (each interacting pair has an area of 10 µm^2^), mating with each other as in the case of a plane air capacitor. Using this approach, it is possible to determine the capacitance of the capacitor by knowing the voltage applied to the plates. Then, with the knowledge of the voltage and capacitance of the capacitor, the charge that accumulates on the plates can be determined.

In order to reduce the computation time, a simplified model of the detection system was created in the CoventorWare software version 10.3 by Lam Research Company, Fremont, CA, USA. During the simulation, it was assumed that the system works under laboratory conditions (stable temperature and humidity conditions). The results are presented in Table 1.

In the case of an accelerometer, the inertia force causes the displacement of the movable plates (along with the seismic mass). By assuming an appropriate ratio of the seismic mass and the stiffness of its elastic suspension, different measurement ranges of the accelerometer can be obtained. Nevertheless, it is the maximal distance between the capacitor plates that is crucial in further analyses.

The behavior of the designed structure of the sensor was simulated under the application of an external load in the form of an acceleration. The applied exemplary accelerations versus the respective displacements are presented in Table 2.

The magnitude of the obtained displacement is dependent on the stiffness of the elastic suspension supporting the seismic mass and its magnitude. Thus, it is necessary to find an optimal geometry and stiffness of the suspension in order to ensure displacements being as large as possible.

The maximal displacement of the seismic mass (corresponding to the full measurement range) is very small. For example, referring to the exemplary mechanical structure by Analog Devices Inc., the maximal displacement is in the order of 1 nm for the ADXL 202E accelerometer, which is about 0.07% of the air gap between plates P1 and P2 (1.4 μm) [46]. Similarly, in our case, it was not possible to locate stationary electrodes E2, E3, and E4 next to each other (as shown in Figure 1 and Figure 2). So, the seismic mass was equipped with three moving electrodes (Figure 6).

The stationary electrodes are arranged in such a way that they align with the appropriate moving electrode at the location corresponding to the predetermined value of the acting acceleration (Figure 7). Thus, each tunneling current detector consists of a pair of dedicated electrodes: one moving electrode and its stationary counterpart (marked in red in Figure 7).

In Figure 7, the sensor is not subjected to any acceleration. Thus, the central pair of electrodes is aligned, which corresponds to a zero value of the acting acceleration, while the other two are misaligned. Should an acceleration with a magnitude equal to ±50% of the measurement range act on the sensor, either the left or the right pair of the electrodes will become aligned, respectively.

### 5.2. Simulations of the Mechanical Behavior of the Seismic Mass

By using appropriate geometric dependencies and assuming a constant initial distance between the pointed electrodes, the geometric change in the distance at which the pointed electrodes will interact with each other was determined. The results of the calculations are shown in Figure 8a,b.

The distribution of points in the presented graphs strongly depends on the initially assumed distance between the tips at the positions where they face each other. The bigger the initially assumed distance between them, the more flattened the region in the vicinity of the point where the tips face each other, i.e., in the region where a highly variable dependence is desired in order to precisely determine the maximal (peak) value of the tunneling current and thus activate the auto-calibration procedure at the appropriate moment, when the expected acceleration acts.

Reducing the distance between the tips significantly improves the shape of the curves, as can be seen in Figure 8b. Unfortunately, a flattened region always exists in the curves. Due to the fact that the distances between the tips will be very small in any real system—about 1 nm, which is the condition for the occurrence of the tunnel current—the curves will be steep enough over their entire length, and there will only be a small area of insensitivity.

Nevertheless, one solution is to employ an additional algorithm that, based on the points of the slopes of the curve, will determine two straight approximation lines and thus ensure greater measurement accuracy within the discussed area by replacing the geometrically determined distance from the flattened area with the point where the lines intersect (Figure 9). In this way, the inflexion point of the curve, which corresponds to a known value of the measured acceleration, will be determined with higher precision.

### 5.3. Simulations of the Tunneling Current Transducer

In order to verify the described idea of the calibration method, preliminary simulations were performed using the ATLAS software version 5.36.0.R by Silvaco Group, Inc., Santa Clara, CA, USA. Two electrodes in the shape of tips were modeled (Figure 10). One of them was made out of aluminum (Al), and the other was made of N-doped silicon with a carrier concentration of 10^16^ cm^−3^. The distance between the tips was 1 nm. The applied voltages were 5 V and 10 V (Figure 10). The model described by Zaidman [48] was used to describe the model of the media transport by the vacuum between the Si and Al electrodes (metal).

The performed stimulations resulted in a graph showing the passage of tunnel current between the electrodes (Figure 11).

As can be seen in Figure 11, within the range of the applied voltages of −5 and 5 V, the field is too small, so no tunnel current is generated. However, in the case of voltages whose absolute value exceeds 5 V, the field is large enough to generate a tunnel current.

The potential and electron concentration distributions for individual cases are presented in Figure 12 and Figure 13.

As mentioned above, the obtained results are based on the Zaidman model. However, recent developments related to tunneling current transducers provide even better opportunities to obtain more advantageous metrological parameters of the proposed transducer. Nevertheless, a standard fabrication process of MEMS devices may be a barrier since it introduces many restrictions on the spectrum of the applied materials and treatments.

## 6. Discussion

The presented concept of the accelerometer with the auto-calibration feature has not been implemented yet. At the present stage of the work, only the mechanical structure of the accelerometer has been designed, and simulation models have been created. Positive results of the simulations related to the behavior of the mechanical structure and the operation of the tunneling transducer have been obtained. For the time being, further stages of development of the sensor are restricted due to the limited opportunities to apply for relatively large financial support.

If the proposed concept were to be implemented in a particular design of a MEMS accelerometer, some problems might occur. They are briefly discussed below.

### 6.1. Amplitude Attenuation and Phase-Shift over Frequency

It is well known that one of the basic shortcomings of MEMS sensors, as far as their dynamic operation is concerned, is the attenuation of the amplitude and the phase-shift of the output signal over frequency [49]. Therefore, in the case of operation under dynamic or even quasi-static conditions, e.g., measurements when the accelerometer is subjected to vibration, the proposed auto-calibration method will be affected by both phenomena. So, for a given type of accelerometer, the permissible frequency at which the auto-calibration ceases to improve the measurement accuracy must be determined. In the case of higher frequencies, auto-calibration should not be activated unless the accelerometer is equipped with special circuits that compensate for these dynamic errors (with a systematic character).

### 6.2. The Thermal Stability of the Stiffness of the Elastic Suspension

The thermal compensation of mechanical structural components has been a well-known problem for a long time in the case of clockworks for watches and clocks. High-precision mechanical regulators (a pendulum or balance wheel with a hairspring) are designed in such a way that either the thermal changes of the dimensions of the pendulum do not affect its total length or the thermal changes in the dimensions of the balance wheel and changes in the elastic modulus of the hairspring compensate each other, keeping the oscillation frequency of the regulator stable. Even though such compensation is mostly achieved through the appropriate selection of the materials used (sometimes even created especially for this purpose, e.g., Invar, Elinvar, and Nivarox), it should be noted that silicon has been successfully applied as the structural material both for the balance wheel and the hairspring, e.g., in some watches by the renowned brand of Patek Philippe [50].

In the case of MEMS devices, the material selection is usually very limited; however, it is easy to manufacture quite complicated shapes of the structural members provided that they are planar [4]. So, it seems that designing a thermally compensated elastic suspension of the seismic mass is an achievable goal.

### 6.3. Tunneling Current Transducers

The proposed detectors employing tunneling current phenomena are not typical. As mentioned above, referring to MEMS accelerometers, usually only one electrode has a tip, whereas the other is a plate [40,41,42,43,44]. However, such a configuration cannot be applied in the proposed detectors as the input quantity in this case is not the distance between the electrodes (which is generally constant), but their alignment. Nevertheless, the proposed original configuration of the pointed electrodes has been simulated and yielded positive results.

### 6.4. Multiaxial Accelerometers

The small maximal displacement of the seismic mass, described as a problem in Section 5.1, is, in contrast, a convenient feature since the detectors, modified as proposed, will not be significantly affected by acceleration, acting in crosswise directions. Only the distance between the electrodes will be slightly changed; however, it will still be their alignment that generates the peak value of the tunneling current (the absolute value of the current is not necessary for detection). So, the proposed design, modified as described in Section 5.1, can be employed for accelerometers with a mechanical structure similar to the ADXL 202E dual-axis accelerometer [51]. Even though it would probably pose a bigger challenge, it can be envisaged that even a triaxial accelerometer can also be modified as proposed.

### 6.5. Alternative Versions of the Accelerometer

The design of the accelerometer proposed in this study is not the only one possible. A few other reasonable and significantly different designs have also been proposed. They are briefly described in the following paragraphs.

#### 6.5.1. A Higher Number of Detectors

To increase the probability that the seismic mass often occupies a position suitable for auto-calibration, more detectors of its specific positions (four or more) may be created, arrayed over the measurement range at different points, including the center position. The use of a larger number of fixed and movable detection electrodes improves the accuracy and efficiency of the calibration.

Moreover, in order to maintain a symmetric structure of the accelerometer (Figure 5), it would be convenient to fabricate two sets of detection detectors: lower ones (Figure 1) and upper ones (on the opposite side).

#### 6.5.2. A Lower Number of Detectors

On the other hand, the number of fixed detection electrodes can be reduced to two or even one, forming two or one detectors with the moving detection electrode for detecting a specific position of the seismic mass. This may be required, e.g., for manufacturing reasons.

#### 6.5.3. Different Locations of Detectors

The proposed specific outer positions of the seismic mass correspond to the value of 50% of the measurement range of the accelerometer. However, this value can be different. Reasonable values seem to be between 10 and 100% of the measurement range. The higher the value, the better the precision of the auto-calibration, but in general, the lower the probability of the seismic mass to be displaced that much.

#### 6.5.4. Different Kinds of Detectors

The proposed idea of self-calibration is not limited to the implementation of the suggested tunneling current detectors, which are rarely used in MEMS. So, it is possible to apply some other detection techniques (e.g., a capacitive accelerometer). However, the accepted technique must feature significantly higher accuracy (yet within a small displacement range) compared to the main measurement principle.

### 6.6. The Fabrication Cost of the Accelerometer

The proposed type of mechanical structure of the MEMS accelerometer is simple and thus ensures low fabrication costs. However, the fabrication of the detectors employing tunneling current phenomena will be more expensive. Unfortunately, at this stage of the work, it is very difficult to evaluate the total fabrication cost of such a sensor die.

## 7. Summary

The proposed precise auto-calibration method can result in a highly accurate MEMS accelerometer despite the existing disturbances and typical shortcomings of MEMS devices. Thus, an original design of a respective MEMS accelerometer has been proposed. An improved performance is achieved through the repeatable activation of the auto-calibration function, employing special detectors of specific positions of the seismic mass.

In the case of mass production, which is characteristic for microsystems, the proposed modifications to the standard design of the accelerometer will slightly increase the fabrication cost, but a significant improvement in the accuracy of its indications will be achieved. Moreover, it will also be possible to eliminate some electronic circuits of the accelerometer (e.g., compensation circuits for temperature errors or aging effects). So, in the case of launching the production of a completely new type of MEMS accelerometer, the total fabrication cost may even be reduced provided that a cost-effective fabrication method for tunneling current detectors is developed at the same time.

In the case of applications where, for reasons of accuracy, the sensor must be cyclically calibrated, the related costs will be much lower compared to typical methods of performing this procedure.

Implementing improvements to MEMS sensors, such as the proposed auto-calibration function, may become very strategic, especially for applications where human life and safety are at stake, e.g., new motorcycle control systems [52] that employ MEMS accelerometers and gyroscopes or diving instruments that include MEMS pressure sensors and accelerometers [1]. Besides this, a sensor with better performance always provides new prospects for its application, especially when its cost is low. Optical measurement systems are one such good example of this idea, e.g., the spectral technique applied in veterinary science [53].

Even though the presented study concerns a MEMS accelerometer, its novel content, which consists of implementing pairs of detection electrodes enabling self-calibration, can be introduced in any sensor, including physical, chemical, or biological sensors, provided that the measured quantity is converted into physical displacement at some stage when generating the output signal. The detection electrodes must not necessarily employ the tunneling current phenomenon: any suitable measurement technique can be applied. The simplest examples of potential implementations of our concept are the aforementioned pressure sensors and gyroscopes or force and displacement sensors.

## 8. Patents

As a result of the works related to the discussed content, the patent in [54] was granted by the Polish Patent Office.

## Figures and Tables

**Figure 1 sensors-24-04018-f001:**
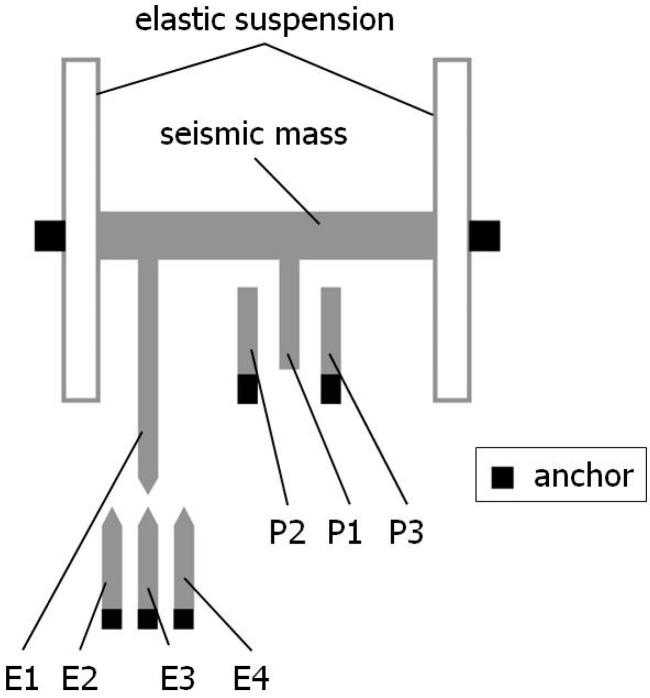
A schematic diagram of the mechanical structure of an accelerometer (at the central position of the seismic mass, *a* = 0).

**Figure 2 sensors-24-04018-f002:**
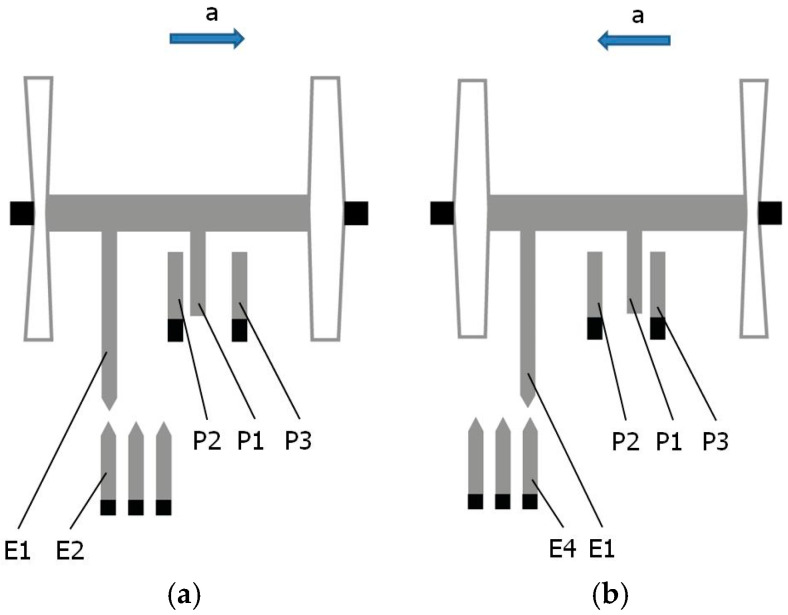
The seismic mass occupies the specific outer position under the action of acceleration: (**a**) the left position of the seismic mass, *a* > 0; (**b**) the right position of the seismic mass, *a* < 0.

**Figure 3 sensors-24-04018-f003:**
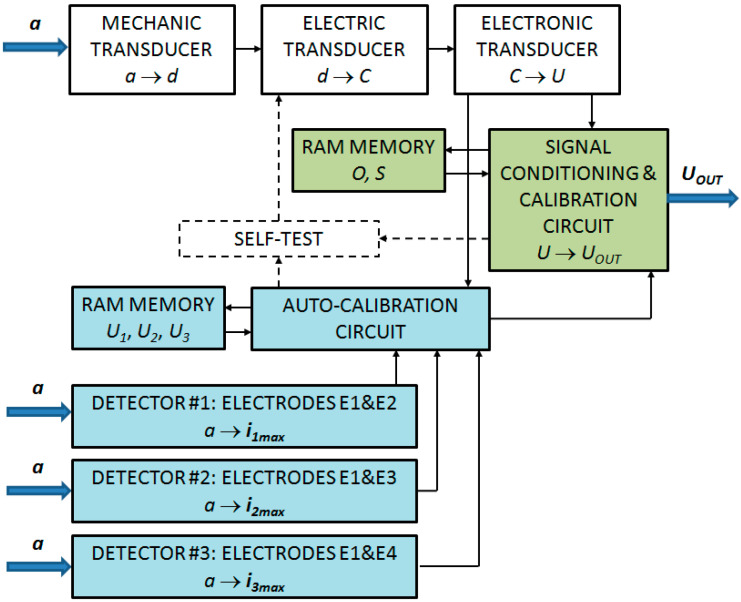
A block diagram of an analog MEMS accelerometer with an auto-calibration feature (color code: blue—new components; green—modified components; white—standard components); *a*—linear acceleration; *d*—the linear displacement of the seismic mass; *d_i_*—the linear position detected by the detector, Di; *C*—the electric capacity, *U*—the raw voltage; *U_out_*—the voltage output signal; *U_i_*—the raw voltage signal generated by the electronic transducer corresponding to position *d_i_*; *i_imax_*—the maximal current signal generated by the detector of the specific position, Di; *O*—the offset; *S*—the scale factor.

**Figure 4 sensors-24-04018-f004:**
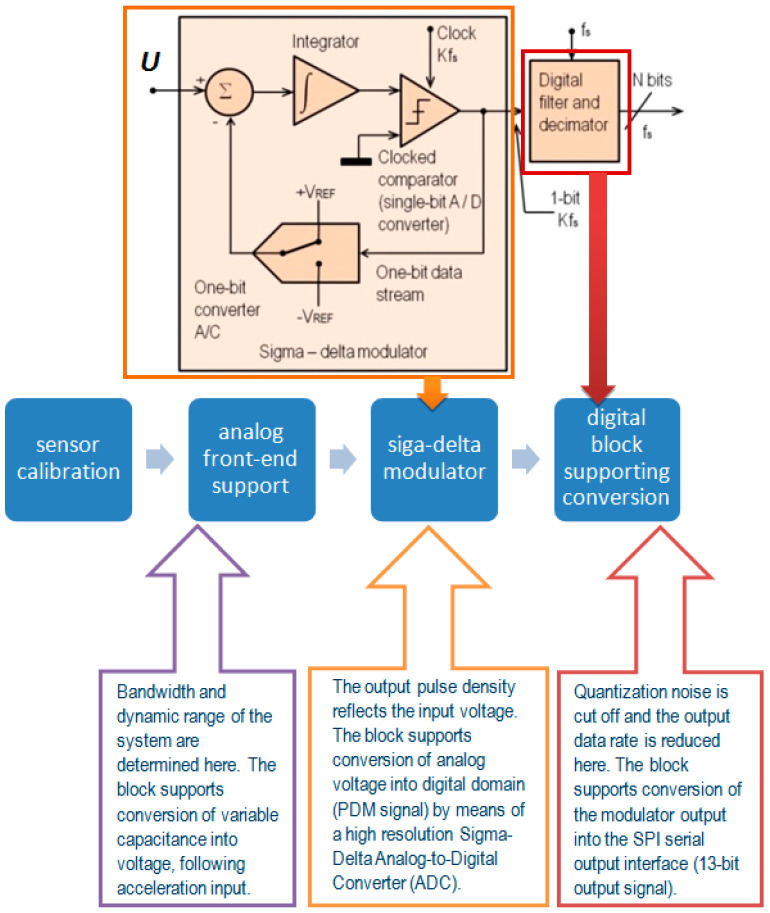
Signal conditioning circuit.

**Figure 5 sensors-24-04018-f005:**
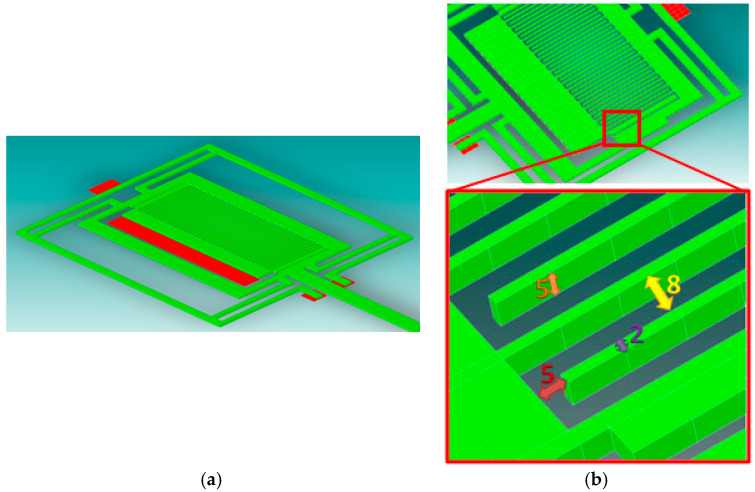
The standard structure of the sensor (without the detection electrodes): (**a**) a 3D model; (**b**) the basic dimensions of the designed structure.

**Figure 6 sensors-24-04018-f006:**
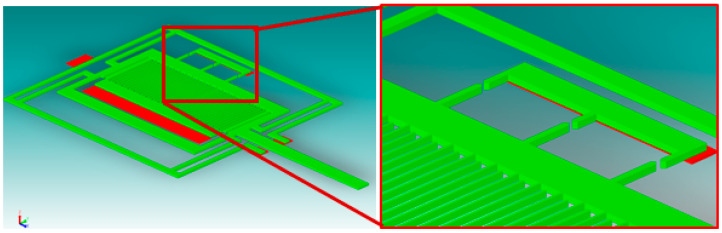
The structure of the sensor with three pairs of detection electrodes.

**Figure 7 sensors-24-04018-f007:**
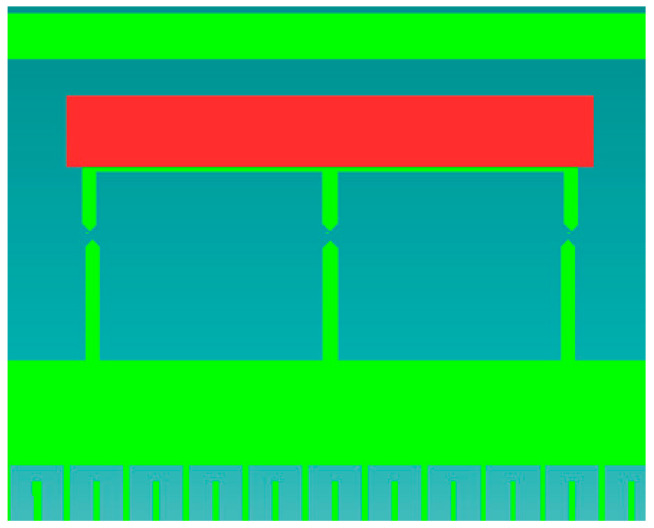
Three pairs of tunneling current electrodes.

**Figure 8 sensors-24-04018-f008:**
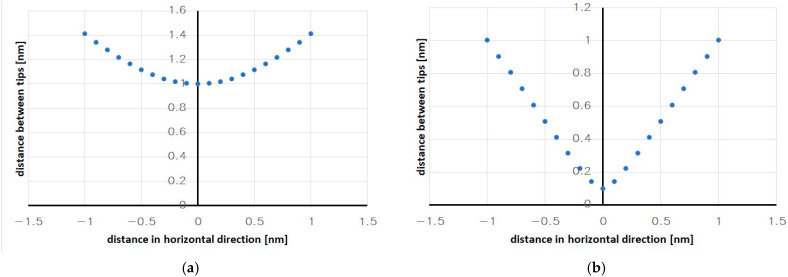
The geometric dependence of the distance between the moving tips of the electrodes: (**a**) a minimal distance of 1 nm; (**b**) a minimal distance of 0.1 nm.

**Figure 9 sensors-24-04018-f009:**
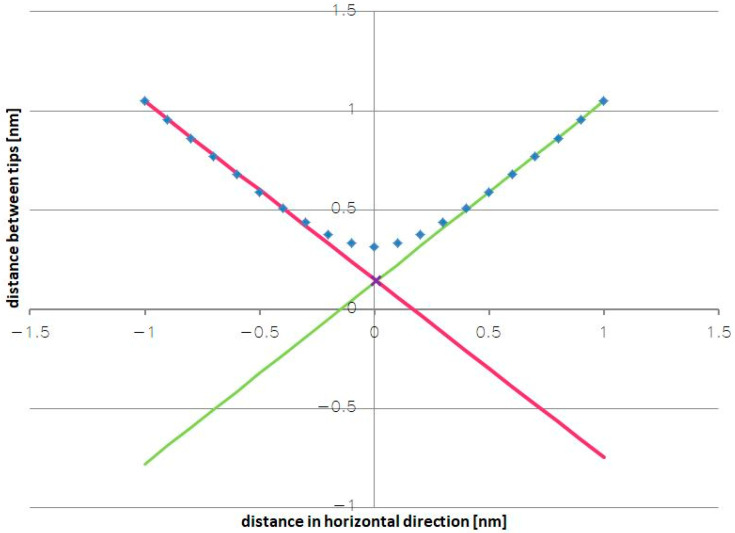
The geometric dependence of the distance between the moving tips (the straight lines illustrate the idea of a linear approximation).

**Figure 10 sensors-24-04018-f010:**
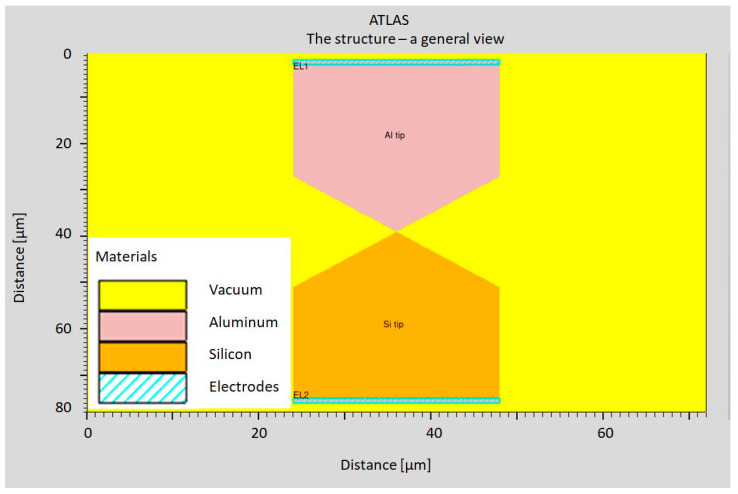
The modeled structure of the tunneling current transducer.

**Figure 11 sensors-24-04018-f011:**
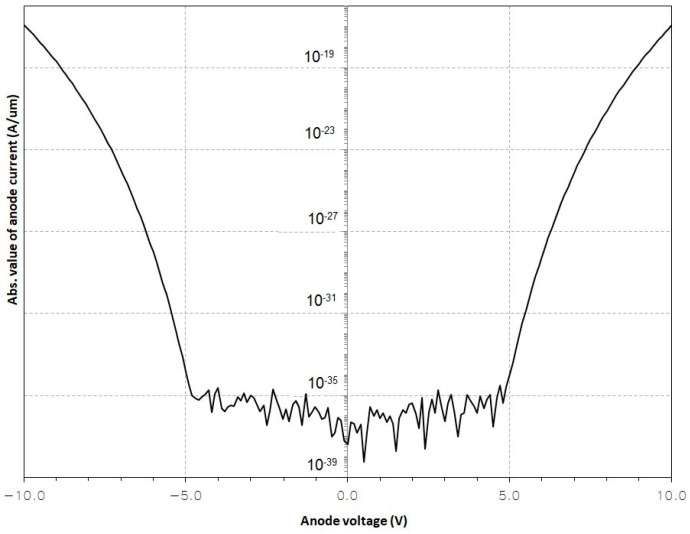
The absorbed value of the anode current.

**Figure 12 sensors-24-04018-f012:**
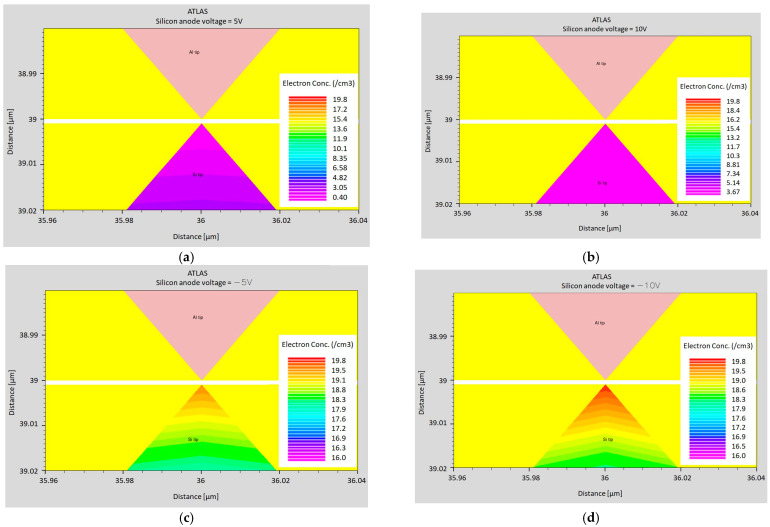
The electron concentration at a distance of 1 nm for the following voltages applied to the silicon anode: (**a**) 5 V; (**b**) 10 V; (**c**); −5 V; and (**d**) −10 V.

**Figure 13 sensors-24-04018-f013:**
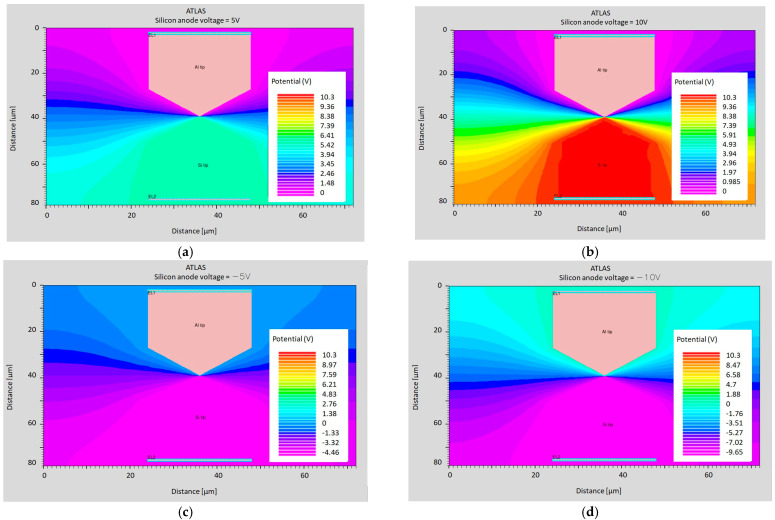
Potential at a distance of 1 nm for the following voltages applied to the silicon anode: (**a**) 5 V; (**b**) 10 V; (**c**) −5 V; (**d**) and −10 V.

**Table 1 sensors-24-04018-t001:** The distance between the capacitor plates versus the achieved charge depending on the voltage applied between the plates; the geometric parameters of the structure, as described in the text.

Voltage	Maximal Distance	Charge
[V]	[nm]	[fC]
20	2.50	0.354
15	1.40	0.266
14	1.20	0.248
13	1.00	0.23
10	0.62	0.177

**Table 2 sensors-24-04018-t002:** The displacement of the seismic mass under acceleration.

Acceleration	Displacement
[m/s^2^]	[nm]
0.5	0.054
1	0.120
1.5	0.160
2	0.220

## Data Availability

The simulation data are available upon request.

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
