# Peer review of "A Method of Precise Auto-Calibration in a Micro-Electro-Mechanical System Accelerometer"

_sensors, 2024, doi:10.3390/s24124018_

Round 1

Reviewer 1 Report (New Reviewer)

Comments and Suggestions for Authors

The article presents a new design of a MEMS capacitive accelerometer fabricated using surface micromachining technology, which has a structure that is capable of accurate self-calibration during operation.

1.    The first line of lines 200 and 204 is indented by two characters.

2.    In the second part, a complete diagram or schematic of the device structure is expected.

3.    All images labelled with (a) and (b) should be in the same format, e.g. Figures 2,4 and 5.

4.    Data in charts 1 and 2 should be centred.

5.    In 5.1, we expect a more specific formulation of the detection electrodes (tunneling current transducers) in the structure.

6.    The third part of the theory needs supplemented. We expect more formulas for accelerometers, tunneling current transducers and calibration.

7.    The article needs a more specific and detailed description of the structure and mechanism of self-calibration.

Comments on the Quality of English Language

Moderate editing of English language required.

Author Response

Dear Reviewer,

First, we would like to thank you for your interesting comments and questions, which enabled us to significantly improve the manuscript.

Generally, we have developed more detailed model of the accelerometer: the signal conditioning circuit (Figure 4) and the mechanical structure including the pairs of the detection electrodes (Figure 6, 7).

All the changes in the manuscript are highlighted by marking with yellow color.

Please find our answers to your valuable comments below.

On the behalf of the authors team, yours sincerely,

Sergiusz Łuczak

Comments of Reviewer #1

  1. The first line of lines 200 and 204 is indented by two characters.
  2. In the second part, a complete diagram or schematic of the device structure is expected.
  3. All images labelled with (a) and (b) should be in the same format, e.g. Figures 2,4 and 5.
  4. Data in charts 1 and 2 should be centred.
  5. In 5.1, we expect a more specific formulation of the detection electrodes (tunneling current transducers) in the structure.
  6. The third part of the theory needs supplemented. We expect more formulas for accelerometers, tunneling current transducers and calibration.
  7. The article needs a more specific and detailed description of the structure and mechanism of self-calibration.

Responses to Reviewer #1

  1. corrected
  2. a new diagram is presented in Figure 4 along with the related description.
  3. we unified all the figures
  4. corrected
  5. the detection electrodes are presented in Figure 6 and 7 along with the related description (Section 5.1); please note that a more accurate representation of the shape of the electrodes in Section 5.3 will not affect the results based on the used simplified model, however it will increase the computation time.
  1. Please note that even at the present form the article is quite long. We wanted to focus on the novelty of our research (i.e. introduction of the pairs of the detection electrodes), thus we decided not to provide a detailed description of the standard members of the sensor (formulas related to mechanical behavior, electric parameters - mainly capacitance, as well as operation of the signal conditioning circuit). With regard to the tunneling current transducers, we fully depended on the model proposed by Zaidman [49] and implemented it in the ATLAS (Silvaco) software. With regard to the calibration, we believe we managed to propose a very simple procedure described by Eq. (2)-(6).
  2. We have added Figure 6 and 7 along with the related description. Besides, we also added a short explanation about the self-calibration at the beginning of in Section 3.

Reviewer 2 Report (New Reviewer)

Comments and Suggestions for Authors

IN this study, the authors describe a unique MEMS capacitive accelerometer design that focuses on perfect auto-calibration during operation, solving accuracy difficulties common in traditional sensors. By adding a mobile detection electrode and three fixed electrodes, the device allows for auto-calibration by sensing precise places of the seismic mass. Simulation investigations validate mechanical and electrical phenomena, and the results reveal enhanced accuracy over standard sensors.

The manuscript is designed well, and it is an interesting study, but some issues should be addressed to improve its quality as suggested below,

        I.            The introduction should carry the necessary information regarding the need of this study and its novelty as a lot of studies had been reported previously.

     II.            Although the suggested adjustments could increase accuracy, the study ignores the difficulties that would arise from incorporating these adjustments into the current manufacturing processes.

  III.            Authors provided very little information about the tunneling current (in section 6.4), but not with valid reasons and references So they should study carefully about the tunneling current as discussed in this recent reports and cite it properly “Low‐Power Negative‐Differential‐Resistance Device for Sensing the Selective Protein via Supporter Molecule Engineering”.

  IV.            More importantly, Can this proposed system be used to fabricate the quantum sensors or biosensors which can be implemented for practical applications?

    V.            The stability of the device should also be verified with electric and optical measurements.

  VI.            The introduction should contain some recently reported sensors as reference.

Remarks: Major Revision is Required to improve the manuscript quality

Comments on the Quality of English Language

minor changes

Author Response

Dear Reviewer,

First, we would like to thank you for your interesting comments and questions, which enabled us to significantly improve the manuscript.

Generally, we have developed more detailed model of the accelerometer: the signal conditioning circuit (Figure 4) and the mechanical structure including the pairs of the detection electrodes (Figure 6, 7).

All the changes in the manuscript are highlighted by marking with yellow color.

Please find our answers to your valuable comments below.

On the behalf of the authors team, yours sincerely,

Sergiusz Łuczak

Comments of Reviewer #2

        I.            The introduction should carry the necessary information regarding the need of this study and its novelty as a lot of studies had been reported previously.
     II.            Although the suggested adjustments could increase accuracy, the study ignores the difficulties that would arise from incorporating these adjustments into the current manufacturing processes. 
  III.            Authors provided very little information about the tunneling current (in section 6.4), but not with valid reasons and references So they should study carefully about the tunneling current as discussed in this recent reports and cite it properly “Low‐Power Negative‐Differential‐Resistance Device for Sensing the Selective Protein via Supporter Molecule Engineering"
 IV.            More importantly, Can this proposed system be used to fabricate the quantum sensors or biosensors which can be implemented for practical applications? 
    V.            The stability of the device should also be verified with electric and optical measurements. 
  VI.            The introduction should contain some recently reported sensors as reference.

Responses to Reviewer #2

I. We added a related paragraph in the Introduction
II. Please note that adding additional teeth (detection electrodes) - on the other side of the mechanical structure of the sensor - to the manufacturing system used for fabrication of MEMS accelerometers will not significantly increase the costs, neither will be difficult to be applied from the technological point of view. 
III. Please note that the mentioned publication is related to the tunneling current meant as a quantum mechanical phenomenon, whereas we use tunneling current transducers employing a small airgap (<1 nm) between two conductors (just as in the case of a scanning tunneling microscope: between the sample and the tip). Nevertheless, we added a short statement at the end of section 5.3 referring to novel solutions of tunneling current transducers.
IV. Thank you for this valuable comment. Even though our design is not related to quantum sensors, we have added in the last paragraph of the Summary a short statement about possible implementation of our calibration concept in the case of other types of sensors. Unfortunately, the concept cannot be applied in biosensors since the mechanical structure (silicon sensor-chip) must be contained within a hermitic packaging.
V. Unfortunately, at this stage we are not able to build a physical prototype of the whole sensor and test its operation experimentally. However, since the proposed concept of the accelerometer has been recently granted a national patent (November 2023), we concluded that presenting our concept to the scientific community might be useful for some researchers. In order to perform relevant experiments we plan to apply for a grant providing necessary financing. However, a successful grant application strongly depends on publication activity related to the grant topic, and this is the main reason why we decided to publish the works realized hitherto, even though they are only at an initial stage.  
VI. We added a related paragraph in the Introduction

Round 2

Reviewer 1 Report (New Reviewer)

Comments and Suggestions for Authors

The article presents a new design of a MEMS capacitive accelerometer fabricated using surface micromachining technology, which has a structure that is capable of accurate self-calibration during operation.The revised article adds a description of tunnelling electrodes to section 5.1.The method of self-calibration of the accelerometers in the article is explained more clearly: the position of the sensitive structure of the accelerometers is confirmed by tunnelling the electrodes in order to correct the output values of the detection combs.

1.The article proposes a method to correct the accelerometer by detecting the signal from the tunnelling electrodes, but it is clear that with the accuracy of the MEMS process, the theory is extremely difficult to apply in practice. The author could follow this idea and think about whether there could be a structure more in line with the MEMS process theory to replace the tunnelling electrodes.

2.As stated in Opinion 1, the title of the article could be amended to ‘Method for Automatic Calibration of Accelerometers’.

3.Optimise the article layout to avoid separating images from icons, as shown in Figure 12.

Comments on the Quality of English Language

The English should be further polished, some sentences should be written in a better format.

Author Response

Dear Reviewer,

We would like to thank you for your positive feedback and apt comments.
All the changes in the manuscript are highlighted by marking with yellow color.
Please find our answers to your valuable comments below.

On the behalf of the authors team, yours sincerely,
Sergiusz Łuczak

Comments of Reviewer #2
1. The article proposes a method to correct the accelerometer by detecting the signal from the tunnelling electrodes, but it is clear that with the accuracy of the MEMS process, the theory is extremely difficult to apply in practice. The author could follow this idea and think about whether there could be a structure more in line with the MEMS process theory to replace the tunnelling electrodes.
2. As stated in Opinion 1, the title of the article could be amended to ‘Method for Automatic Calibration of Accelerometers’.
3. Optimise the article layout to avoid separating images from icons, as shown in Figure 12.
4. Comments on the Quality of English Language: The English should be further polished, some sentences should be written in a better format.

Responses to Reviewer #2
1. Please note that the proposed accelerometer structure is a typical comb drive commonly used in the case of these sensors. As for the accuracy of fabricating a planar structure, e.g., using electron-beam lithography, there is rather no risk of a failure. The only problem may be the vertical dimension, i.e. the thickness. It would be probably necessary to use a SOI wafer with a pre-selected thickness of the tool layer and use both techniques for making MEMS as well as CMOS when fabricating the structure.
Please note that e.g. in references [41-44] there are examples of employing a tunneling current transducers in the structure of a MEMS accelerometer. Nevertheless, in Section 6.5.4 we suggested implementation of detection techniques other than application of the tunneling electrodes.
2. According to the suggestion, we have modified the title, which reads now: "Method of Precise Auto-calibration in MEMS Accelerometer"
3. We have changed the layout.
4. We have introduced some lingual improvements over the whole text.

Reviewer 2 Report (New Reviewer)

Comments and Suggestions for Authors

Accept it in the present form

Author Response

Dear Reviewer,

We would like to thank you again for your positive feedback and apt comments.
All the changes in the manuscript are highlighted by marking with yellow color.

On the behalf of the authors team, yours sincerely,
Sergiusz Łuczak

This manuscript is a resubmission of an earlier submission. The following is a list of the peer review reports and author responses from that submission.

Round 1

Reviewer 1 Report

Comments and Suggestions for Authors

Comments:

  • Introduction – paragraph about the organization of the paper is missing
  • Introduction – the state-of-the-art (SOTA) is elaborated on general level. The main contributions of the article are defined, but they should be more highlighted – the using of items is highly recommended. Next, author has a bit higher self-citation oneself – only the most relevant and important (own) works should be cited. Finally, a comparison table should be very helpful for readers to check the features of the proposed solution and existing ones.
  • Section 2 – there is no reason to present Figs. 2 and 3 separately. I recommend for the authors to merge Figs. 2 and 3 into one figure with subfigures a) and b).
  • Section 2 – the main difference between the standard and modified accelerometer is that the standard electric structure had been enhanced by adding four detection electrodes?
  • Section 3 – the readability of “formulas” should be improved. Please, make these items more visible! Thanks!
  • Section 4 – the structure of the accelerometer is discussed on very general level and mainly in brief. I understand that the proposed concept is under evaluation as a patent. On the other hand, without discussion of the main parts of the analog MEMS accelerometer with auto-calibration in detail is hard to decide about its novelty (and main contributions).
  • Section 5 – it is written: “The presented concept of the accelerometer with the auto-calibration feature has not been implemented so far, due to relatively high fabrication costs of a MEMS sensor.” I understand, but how was its functionality tested? Are there available any simulation-based data?     
  • Section 5 – there are not available any simulation (or measurement)-based evaluation of the proposed MEMS accelerometer?

Reviewer 2 Report

Comments and Suggestions for Authors

The paper presents a concept of a MEMS accelerometer including an auto-calibration capability. Even the paper is clear written is described only an idea with possible benefits related to other solutions.  The terms stated in the title "precise auto-calibration.." should be demonstrated, what means in this case "precise" with respect to the accuracy of the device?  I consider that a validation of the concept is absolutely necessary. On other hand the algorithm is based on predetermined acceleration of a strict value (e.g. equal to 50% of the measurement range) that means it relies on initial calibration of the accelerometer done by other methods. How this will influence your concept / accuracy?